# An Industrial Case for Polypropylene Nanocomposite Foams: Lightweight, Soundproof Exterior Automotive Parts

**DOI:** 10.3390/polym14061192

**Published:** 2022-03-16

**Authors:** Burcu Girginer Ozunlu, Fatma Seniha Guner

**Affiliations:** 1Material Science and Engineering Department, Graduate School of Science Engineering and Technology, Istanbul Technical University, Istanbul 34467, Turkey; girginer@itu.edu.tr; 2Farplas Automotive R&D Center, Cayirova 41420, Turkey; 3Department of Chemical Engineering, Istanbul Techical University, Istanbul 34467, Turkey

**Keywords:** foam injection molding, graphene nanoplatelet, automotive industry, polypropylene

## Abstract

Lightweighting is a challenge for the automotive industry, and foaming is a key technology used to address this problem. A new practical approach is studied to regulate the cell formation of copolymer polypropylene (co-PP) by utilizing graphene nanoplatelets (xGnP) as a process aid during foam injection molding. The approach was designed to enable process freedom to tune part performance by adjusting the amount of xGnP masterbatch. Two different levels of 1–2 wt % xGnP and 0.25–0.35 wt % supercritical fluid (SCF) were investigated. Prepared samples were compared with samples prepared by the traditional method (twin-screw extrusion followed by foam injection molding). The nanocomposite with 2 wt % xGnP comparatively showed about twofold reduction in cell size magnitude. Although the increment in SCF amount resulted in a 47% and 122% enhancement in flexural modulus and strength, respectively, and a 45% loss in Izod unnotched impact strength, the cell size was prone to increasing with regard to low melt strength as compared to neat foams. In conclusion, a 12% weight reduction fulfilled the desired performance parameters in terms of mechanical and sound insulation by utilizing 2 wt % xGnP as a process aid.

## 1. Introduction

The automotive industry is paying great attention to lightweighting to meet the requirements against carbon dioxide emissions. Foam injection molding is a practical method for manufacturing plastic trim parts in an ecofriendly manner. The amount of plastic trim is supposed to be approximately 13% of the total weight of a vehicle [1]. Unfilled heterophasic polypropylene copolymer (co-PP) is a type of cost-efficient, impact-modified copolymer that is widely used in exterior unaesthetic trim parts. Engine covers, trims, and luggage components are applications with co-PP, and their main functions are as coverings. Thus, flexural and impact behavior are more important mechanical properties of these materials than the tensile property is due to the main functionality of this application. Additionally, sound insulation is a desired function.

However, co-PP foams exhibit a nonuniform cell size and distribution, resulting in deteriorated mechanical properties [2]. This behavior occurs because of the low melt strength, poor gas solubility, and two-phased morphological structure of the material. Thus, the ultimate weight reduction attained in practice is below 8%, considering processing problems at the industrial scale such as short shot or postblowing [3]. Although numerous studies have been reported in the literature on the foaming of polypropylene materials, few focused on the relations among material structure, process parameters, and part performance. Investigations were mostly carried out on lab-scale batches or semicontinuous processes throughout the standard test bars or plates, and consequently provide limited insights. However, most automotive trim parts have a complex design consisting of elements such as bosses, ribs, and holes. Moreover, the high shear rate plays a dominant role in industrial-scale processing, which has a strong influence on foam formation. Frictional forces, loss on the pressure and melt temperature become much more significant through the long flow path from the gate to the endpoint in the mold of complex structures. Industrial foam samples have a macrocellular cell structure in the core layer regarding the orientation of additives governed by fountain flow and in efficient cooling [4,5]. To address this problem, there are many studies examining the utilization of nanoparticles from many perspectives [6,7,8,9,10], and graphene nanoplatelets (xGnP) represent the most promising option among nanoparticles thanks to their scalable production and affordability.

The preparation of xGnP-co-PP nanocomposites is challenging due to agglomeration caused by particle–particle interactions [10]. Moreover, the enhancement of the exfoliation degree of nanoplatelets is a requisite condition to fulfill the utmost improvement of mechanical and other properties. Many attempts to enhance the dispersion and exfoliation of the xGnP in the polymer matrix were reported in the literature, and the utilization of supercritical fluid (SCF) is a promising approach to address this challenge relating to its strong solvent and intercalation role [11,12]. SCF-assisted injection molding studies showed that the properties of thermal and electrical conductivity electromagnetic interference shielding can be improved by exfoliation of platelet structures [13,14,15,16]. In the literature, it was proven that the pretreatment of layered structured nanoparticles with SCF gases under batch conditions resulted in an increase in interlayer gallery spacing [17,18,19]. However, the processing time during which a nanoparticle is subjected to SCF is not sufficient, and shear forces are predominant in continuous systems. There are few studies in the literature concentrating on the foam injection molding process. The cocontinuous concept of foam injection molding, first involving the pretreatment of clay nanoparticles with SCF during twin screw extrusion with low-density polyethylene, and subsequently foam injection molding of a derived nanocomposite as a second step, was investigated. Although ultimate weight reduction levels of about 15% and a decrease in average cell size were attained, the concept has drawbacks. The shelf-life limitations of the first step eventuated in an additional storage unit of semiproducts, and the aforementioned steps are not collocational operations in one single manufacturing site [20]. The dispersion and exfoliation states of graphene nanoplatelets were investigated in a homopolymer polypropylene matrix by using subcritical gas-assisted processing during melt blending [21]. The expansion of bubbles provides equibiaxial flow during cell growth. A dominant extensional flow driven by bubbles had a dominant effect on agglomerates accompanied by shear forces. As with xGnP, the effect reported on clay with SCF provides insights into this phenomenon. The mechanism of the enhancement of the dispersion and exfoliation of xGnP powder treated with SCF before foam injection molding was also reported [22]. SCF gas molecules penetrate through the layers of clays thanks to their high diffusivity. Separation between interlayers occurs due to cell growth during the foaming steps, especially in the case of the subjection of SCF gas in the prefoaming stage. Consequently, exfoliation and dispersion are accompanied by a uniform foam morphology. In another study, xGnP with high-density polyethylene showed similar properties, in agreement with these findings [23].

In this study, a new and facile approach to prepare co-PP foams with enhanced cell morphology by means of the utilization of dispersed and exfoliated xGnP is studied. We hypothesize that providing a practical study for designers and engineers in the automotive industry, choosing a representative exterior trim part design, and performing the experimental studies via industrial-scale equipment fill the gap in the literature from the industrial perspective. Therefore, we aimed to investigate (i) the enhancement of the dispersion and exfoliation degree of xGnP particles driven by the high shear forces of industrial-scale equipment; and (ii) the interaction between the SCF gas and xGnP platelets, and its effect on nanocomposite properties and structure formation during cell growth with the arrangement of xGnP particles through cell growth. Through a series of experiments, we evaluated the incorporation of xGnP into injection molding grade with commercial co-PP. Predispersed xGnP nanoplatelets in masterbatch form dilution were tested via foam injection molding in a one-step process, using 1 and 2 wt % xGnP. Specifically, these experiments are important for evaluating the two levels of SCF at 0.25 and 0.35 wt % with regard to performance to understand the effect on cell formation and dispersion state to compare the findings with the traditional nanocomposite foam preparation approach [3]. Therefore, the foam samples were prepared in a two-step process. First, 1 and 2 wt % xGnP were melt-mixed with co-PP via twin-screw extrusion. Second, nanocomposites were subjected to foam injection molding. To ensure sufficient xGnP–co-PP interaction, maleic anhydride grafted polypropylene was used as a coupling agent. Lastly, we characterized the premixed samples in terms of physical, mechanical, and thermal properties, and evaluated all foam samples through morphological, mechanical, rheological, and structural analyses, and sound transmission loss measurement.

## 2. Materials and Methods

### 2.1. Materials

Heterophasic polypropylene ethylene copolymer (polypropylene-*co*-ethylene, co-PP), with a narrow molecular weight distribution and the trade name of Moplen Hexp 2000P, which is commercially used in exterior trim part production, was provided by Lyondell Basell Technology Company (Frankfurt, Germany). The material had a melt flow index value of 16 g 10 min^−1^ (230 °C, 2.16 kg ISO1133). Graphene nanoplatelets incorporated with a polypropylene masterbatch (xGnP MB), with the trade name of XGPP^TM^, were supplied by XG Science Inc. (Mason, MI, USA). The masterbatch comprised 10% graphene nanoplatelets with an average surface area of 300 m^2^ g^−1^, bulk density of 0.03–0.10 g cm^−3^, thickness of 10–20 nm, and a lateral size of 1–2 µm (data from XG Science Inc.). Maleic anhydride grafted with polypropylene homopolymer (polypropylene-*graft*-(maleic anhydride, PP-g-MAH), with the trade name of Bondyram 1001, was used as a coupling agent (CA) and was provided by Polyram Ram-On Industries L.L (Moshav Ram-on, Israel). Its density was 0.90 g cm^−3^ (ISO 1183) and it had a melt flow index value of 100 g 10 min^−1^ (190 °C, 2.16 kg—ISO 1133). The maleic anhydride content was 1%, which was determined by FT-IR, and the melting point was 160 °C (ASTM D3418) (data from Polyram Ram-On Industries L.L.). Nitrogen gas with 99.9% purity was used as a blowing agent during foaming experiments and had been purchased from Linde Gas (Kocaeli, Turkey).

### 2.2. Foam Injection Molding

Foam injection molding experiments were performed on an industrial-scale injection molding machine with a clamping force of 1600 ton-f (KM-MX1600 Model Krauss Maffei GbmH, München, Germany) and equipped with a Trexel SCF unit (Trexel Inc, Wilmington, NC, USA). The part design representing an automotive exterior trim part such as an engine undercover was used as a test plate in the experimental study. The part design comprised frequently used design elements of mechanical reinforcement and joints, such as ribs, bosses, screw nests, welding dots, grids, joint nests, and flaps, as indicated by 1–6 in Figure 1, respectively. The average wall thickness (z) and part dimensions in width and length (x and y, respectively) of the parts were: z: 2.50 mm and x: 450.00 mm, and y: 975.00 mm (Figure 1). The single cavity mold with a hot runner manifold system and 5 direct gates (without sprue) to prevent preblowing, corresponding to the part design, was manufactured by Farplas Automotive Company (Kocaeli, Turkey).

The nonfoam (solid) and foam samples of neat co-PP and its nanocomposites were prepared by injection molding. xGnP MB was added directly to the foam during injection molding to obtain final concentrations of 1 and 2 wt % in Method 1 (Figure 2a). To compare our experimental results with the conventional approach, corresponding counterparts of nanocomposites comprising 1 and 2 wt % xGnP and 1 wt % coupling agent were prepared via Method 2 (Figure 2b). Method 2 had 2 steps: Step 1 was twin-screw extrusion (TSE), and Step 2 was foam injection molding (Figure 2b). The process parameters were set as the same for Methods 1 and 2, and those of Step 2 are given in Table 1.

The process parameters of TSE were set as follows: PP copolymer (co-PP), 1 wt % PP-g-MAH and 1 or 2 wt % xGnP equivalent masterbatch amount were premixed and then fed into a corotating compounding extruder (Werner & Pfleider GmbH, ZSK 25; Stuttgart, Germany) with screw-diameter (D) and length (L) values of 25 and 48 mm, respectively. The applied cylinder temperature profiles were 150, 155, 160, 170, 180, 190, 195, 200, 205, 210, 215, 220, and 220 °C, and the shear rate was set as 450 rpm. All ingredients were dried at 120 °C for 4 h in moisture- and temperature-controlled environments using drying ovens with high precision according to the instructions given by the material providers [3]. Extrudates were soaked in a water bath for cooling, dried, and pelletized.

The representative test specimens with dimensions of 2.50 × 10 × 80 mm corresponding to the ISO 179 1A standard were punched via a hydraulic press and steel knives from the marked red area (Figure 1) for characterization. Additionally, the disk-shaped samples for rheological measurements with a 25 mm diameter parallel plate were punched from the same area. The test specimens of transmission loss measurements with a 30 mm radius and a thickness of 10 mm (Figure 1) were cut via a computer numerical control (CNC) machine. To guarantee proper dispersion and distribution, and to prevent agglomeration, a screw design was chosen with a configuration that enabled intense shear and high residence time, arranged with kneaded convey blocks and the design of the selected screw configuration [24].

Samples produced via twin-screw extrusion were injection-molded by using an Engel ES 1350/200 HL (Engel GmbH, Isernhagen, Germany) injection-molding machine to produce ISO 527-2 Type 1-A and ISO 178 standard test bars. Processing parameters were given in detail in our previous study [25]. The injection-molded premixed samples are shown in Table 2.

During the foam injection-molding sample preparation procedure, the following criteria were taken into consideration. For samples with a part weight within a tolerance limit of ±10%, theoretical part weights were collected. The foam samples prepared via Methods 1 and 2 are summarized in Table 3.

### 2.3. Characterization

#### 2.3.1. Melt Flow Index

The melt flow index (MFI) test was conducted on pellets of samples (Table 2) and raw materials as received, and using Instron Ceast MF20 (Instron Gmbh, Darmstadt, Germany) under a 230 °C flow temperature by using a base-load EN ISO 1133.

#### 2.3.2. Density

The density of the samples (Table 2 and Table 3) was determined in accordance with ISO 1183 standard. Foam injection-molded samples were weighted in alcohol and air. The void fraction of samples was calculated using the following equation:$$v_{f}=1-\left(\frac{1}{\phi }\right)$$
where *v_f_*, void fraction; ϕ, calculated ratio of density of nonfoamed and foamed samples [6].

#### 2.3.3. Determination of Ash

To determine the weight loss percentage of the samples, standard ash content analyses were conducted following the ISO 3451 method. Approximately 5 g samples were placed in a dried and preweighted porcelain crucible, and burned off at up to 650 °C under atmospheric conditions. Following this, samples were dried in a desiccator at room temperature and weighted to determine their amount of nonpolymeric substances.

#### 2.3.4. Differential Scanning Calorimetry

Differential scanning calorimetry (DSC) measurements of polymers and nanocomposites were carried out under a nitrogen atmosphere from room temperature to 200 °C using Perkin Elmer Diamond 4000 model (Perkin Elmer Gmbh, Rodgau, Germany). Approximately 7 mg of the dried samples was heated and held at 200 °C for 5 min to eliminate thermal history. To determine crystallization and melting properties, second heating and first cooling cycles were performed at a rate of 10 °C min^−1^. The associated thermal parameters of crystallization (Tc) and melting (Tm) temperatures, crystallization enthalpy (ΔHc), and the heat of melting (ΔHm) were extracted using Pyris software.

#### 2.3.5. X-ray Diffraction

X-ray diffraction (XRD) was carried out by using a Bruker AXS Advance Diffractometer with a CuKα radiation source (Brucker, Karlsruhe, Germany). Measurements were performed with the following setting parameters: 2 theta intervals between 5 and 35, step size was 0.02 degrees, and time step was 1 s.

#### 2.3.6. Raman Spectroscopy

Raman spectroscopy was performed by using a Renishaw InVia Reflex Raman Microscope and Spectrometer (Cardiff, UK). The spot size was obtained as being between 1 and 2 µm using a 50 magnification objective lens with a long working distance (LWD). Measurements were taken at 3 different spots along the cross-section of the skin layer.

#### 2.3.7. Heat Deflection and Vicat Softening Temperature

The heat deflection temperature (HDT) of the twin-screw extruded samples (Table 2) was measured by a Zwick Roell HDT/Vicat instrument (ZwickRoell Gmbh & Co., Ulm, Germany) using the ISO 75-B method with 0.45 MPa load. The Vicat softening temperature (VST) was measured following the ISO 306-B standard with a load of 50 N and a heating rate of 50 °C h^−1^.

#### 2.3.8. Mechanical Property Measurements

The mechanical behavior of the samples is shown in Table 2 and Table 3. The Zwick Roell Proline Z020 Universal Testing Machine (ZwickRoell Gmbh & Co., Germany) was used to measure the tensile strength and tensile modulus. All test specimens were conditioned for 48 h. The test speed was set to be 50 mm min^−1^, and the gauge length was set at 105 mm. Furthermore, the flexural modulus and strength values of the samples were determined by ISO 178. The test speed was set to be 5 mm min^−1^.

The Izod impact strength (unnotched) of samples was determined by the Zwick Roell HIT 25 pendulum impact tester (ZwickRoell Gmbh & Co., Germany).

#### 2.3.9. Scanning Electron Microscopy

The morphology of foam samples was examined with the scanning electron microscope SEM JEOL JIB-4601F MultiBeam FIB-SEM system. The specimens were fractured in liquid nitrogen for 3 min and coated with a thin gold layer before SEM observation. To achieve foam morphological numerical analysis, Image J software was used, and calculation was conducted using the formula presented in the literature:$$average\;cell\;density={\left[\frac{nM^{2}}{A}\right]}^{\frac{3}{2}\;}{\left(1-v_{f}\right)}^{-1}$$
where *n*, *M*, *A*, and *υ_f_* are the average number of cells in the micrograph, magnification factor, micrograph area, and void fraction, respectively [26].

#### 2.3.10. Rheological Measurements

Rheological measurements of foam samples were performed via Anton Paar MCR 301 Rheometer (Anton Paar Germany GmbH, Ostfildern, Germany). All experiments were performed in parallel plate geometry with a 25 mm diameter at a temperature of 200 °C under a nitrogen atmosphere.

#### 2.3.11. Sound Transmission Loss Measurement

The sound transmission loss of foam samples was determined by the ISO 10534-2 test method by an impedance tube, BSWA SW 477 model (BSWA Tech, Beijing, China).

## 3. Results and Discussion

The results of measurements corresponding to produced samples were presented in two parts as characterization of semiproducts by twin-screw extrusion (TSE) and foam samples by injection molding.

### 3.1. Characterization of Prepared Semiproducts

#### 3.1.1. Effect of Graphene Nanoplatelet Addition on Physical, Mechanical, and Thermal Properties

Physical, mechanical, and thermal properties of semiproducts were explored in comparison with those of graphene nanoplatelet (xGnP) and polypropylene ethylene copolymer (co-PP) (as received) as raw materials. As the reference point to evaluate the samples prepared by Method 1, we aimed to understand the effect of xGnP amount and twin-screw extrusion mixing method on these aspects (Table 4). According to the results, the density of samples remained almost the same with the addition of xGnP. An increase in the melt flow index (MFI) value was obtained for all semiproducts. Although this increase was expected due to the presence of coupling agent, the MFI value of nanocomposites proportionally increased with the xGnP amount in the presence of the same amount of coupling agent. This result is compatible with the literature because two-dimensional layered nanoadditives, e.g., clay, graphene, and graphite, demonstrate increased mold flow index [27,28]. This behavior is explained as the flow-induced orientation of platelets owing to the high aspect ratio, causing the enhancement of flowability during the MFI test. Additionally, xGnP inherently plays a role as a solid internal lubricant, similar to sliding over chains regarding platelet structure. Contrarily, the addition of graphene platelet nanoparticles had a reducing effect on flowability due to the restriction of chain mobility [7,29].

The ash content of nanocomposites was obtained as 0.68 and 0.89 g for the prepared nanocomposites. Mechanical properties of twin-screw extruded neat co-PP were deteriorated due to possible degradation during processing, as expected. However, the addition of xGnP recovered the loss of properties, except for impact strength behavior. The tensile modulus of nanocomposites with 1 and 2 wt % xGnP showed improvement by 27.25% and 38.27%, respectively, regarding the contribution of nanofiller to stiffness and crystallization compared to their neat counterparts. The magnitude of improvement showed consistency with empirical and theoretical studies regarding graphene-polypropylene nanocomposites in the literature [30,31,32]. The improvement of the modulus for copolymer-based nanocomposites was lower than that of homopolymer-based nanocomposites due to crystallinity degree [33,34]. The flexural property of nanocomposites showed an analogy with tensile properties. The flexural modulus was increased by 24.00% and 27.00% for nanocomposites with 1 and 2 wt % xGnP, whereas increases of 18.40% and 28.50% were observed in flexural strength. The degree of improvement in tensile strength was higher owing to the alignment of nanoplatelets during the tensile test. The remarkable loss of impact strength was observed by the addition of xGnP masterbatch into neat co-PP, agreeing with the literature [32]. This is attributed to the poor impact strength of the masterbatch regarding the homopolymer-based matrix. Secondly, the deterioration of the energy dissipation is governed by increased stiffness. Overall, the impact strength of the prepared nanocomposites meets the requirements of underbody exterior trim parts, which are determined as being above 10.00 kJ m^−2^ at 23 °C by the automotive industry, in practical terms.

Although the presence of the coupling agent has a mitigating effect, the heat deflection temperature (HDT) value of nanocomposites was proportionally increased with the addition of xGnP with respect to the enhanced crystallinity, and agreed with the literature (Table 4) [35,36]. Additionally, the incorporation of xGnP enhances VST depending on the enhancement of the modulus and agreement with data in the literature [37,38]. In the automotive industry, part performance requirements are subject to 80 °C extreme temperature environmental conditions for exterior applications [39]. Therefore, the 2 wt % xGnP incorporated nanocomposite sample is the most promising material to meet the test requirements of the environmental aging test among the developed materials in this study in terms of heat deflection under specific loads.

#### 3.1.2. Effect on Graphene Nanoplatelet on Crystallinity

The mechanical properties and foaming behavior of polypropylene copolymer-based nanocomposite are in close correlation with their microstructure and degree of crystallinity.

The thermal properties of neat samples as received and processed with maleic anhydride grafted with polypropylene homopolymer (PP-g-MAH) and their nanocomposite counterparts were characterized and are displayed in Figure 3a–c. The addition of coupling agent and processing through twin-screw extrusion had no significant effect on the melting and crystallization behavior of neat material. It was assumed that degradation due to processing slightly affects the microstructure, as seen in the literature [40]. Figure 3a shows the crystallization curves of all samples in the first cooling thermogram, and the shift toward higher temperatures in correspondence with the amount of xGnP depending on its nucleating agent role is evident. Crystallization temperature rose by about 5 °C in nanocomposites, and this could be attributed to the sufficient dispersion and distribution of nanoparticles, as reported in our previous study and others in the literature (Table 5) [33,41].

Crystallization enthalpy was similar for all samples, as shown in Table 5. The crystallization half times of nanocomposites were reasonably lower than those of the neat samples (Figure 3b). Figure 3c presents the second melting curves of samples. Characteristic peaks of two dominant contributions in heterophasic endotherms were observed in all samples. The intense peak around 164 °C referred to polypropylene segments, and the smooth peak around 118 °C was related to the secondary contribution of polyethylene or ethylene/propylene copolymer with long ethylene sequences. The shoulder peaks appearing in front of the intense melting peak of polypropylene in co-PP-xGnP1-TSE and co-PP-xGnP2-TSE indicate reorganization or recrystallization of polypropylene’s morphological structure. In other words, the addition of xGnP induced the formation of polymorphism. The contribution of polypropylene chains with different lengths existing in masterbatch resin most probably promotes this polymorphism. Moreover, according to recent studies, graphene nanoplatelets have a β nucleating agent role on iso-polypropylene’s structure. They studied the effect of melting, mixing parameters, screw configuration and the amount of graphene nanoplatelet-reinforced isotactic PP on the nucleation of the β-polymorph [42]. Results indicate an α–β transition with featured broad and asymmetric peaks with secondary shoulder appearance around 165 °C. The observed change in the melting curve is similar to the evidence seen in Figure 3c. From another point of view, the applied high shear to maintain sufficient dispersion may lead to α–β transition.

In brief, the addition of xGnP nanoplatelets enhances flowability, and tensile and flexural behaviour thanks to improved crystalline structure. It does not affect the density of nanocomposites compared to unfilled co-PP. Thermal behavior was improved by the incorporation. Overall, the nanocomposite comprising 2 wt % xGnP met the automotive standard requirements.

### 3.2. Characterization of Foam Molded Samples

#### 3.2.1. Foam Morphology

First, to evaluate the dispersion state of the nanoparticles in the polypropylene copolymer matrix, morphological analysis of a nonfoam (solid) injection molded neat sample and its nanocomposite counterparts was performed via scanning electron microscopy (Figure 4a–e). Second, foam samples were analyzed to understand the effect of xGnP addition on foam morphology (Figure 5). The general characteristic of polypropylene copolymers, having some voids because of the residue of ethylene comonomer in the gas phase from the polymerization step, can be seen for all solid samples (Figure 4a–e) [38]. According to the images, the fracture surface of neat specimens was smoother when compared to their nanocomposite specimens, and the unsmoothed cracked surface faded in the nanocomposites with 2 wt % xGnP. This can be explained by the homogenous reinforcement of neat samples. Discrete xGnP particles were observed and are marked with red arrows regarding poor xGnP particle–matrix interactions, even in the presence of coupling agent, in Figure 4b–e. The enhancement of strength in nanocomposites depends on the interaction between their constituents as well as reinforcement aspects such as size and aspect ratio. Similar free xGnP particles separate from the polymer matrix, relying on the insufficient wetting of the interface between polymer and filler, were recorded in the literature [27].

Agglomerates approximately 40 µm in size were determined for a system comprising the xGnP particles possessing similar dimensions [31]. Thus, large agglomerates, wrinkling, and the buckling of nanoplatelets were not seen due to the relatively low aspect ratio and high shear forces of the industrial-scale equipment in our study. Overall, the morphological investigation supports the mechanical results and agrees with the literature.

Figure 5a–j show the representative foam morphology of the neat polymer and its nanocomposite counterparts prepared by Methods 1 and 2, and being subjected to 0.25 wt % (low) and 0.35 wt % (high) (supercritical fluid) SCF percentages. To quantitatively evaluate average cell size and density, the ratio of skin thickness to foam thickness was extracted from SEM images by using Image J software and is shown in Table 6. Figure 5a,f shows neat polypropylene foams exhibiting copolymeric morphological characteristics of nonuniform cell distribution with polydisperse cell sizes governed by different SCF dissolution affinities in polypropylene and ethylene phases [2].

Additionally, collapsed bubbles were determined as being due to poor melt strength. Likewise, cell sizes of a few hundred elongated microns were determined in neat samples as having been caused by insufficient SCF levels and dominant shear forces, as expected [43]. The incorporation of xGnP provides improved uniformity of cells compared to neat polypropylene copolymer due to its nucleating agent role. Additionally, cells formed in the core layer rather than the skin layer because of the increase in skin thickness–foam thickness ratio. Only the foam samples prepared via Method 1 behaved exceptionally in terms of this ratio. Moreover, collapsed cells were observed in almost all samples owing to the low melt strength, and it seems that the addition of xGnP did not eliminate collapse completely. Only nanocomposites with 2 wt % xGnP subjected to low SCF showed an enhancement of cell size compared to neat polypropylene copolymer among all the samples for both methods (Table 6). The increment in cell size is postulated as being due to the sweep of xGnP nanoplatelets by fountain and shear flow parallel to the skin layers. It seems that the increase in SCF amount might promote this behavior via its contribution to viscosity enhancement, as well as sweep by bubble expansion. Our results agree with studies in the literature [22]. Although a sandwich foam morphology is expected with respect to the gradual concentration of xGnP as stated in the literature, efficient rapid cooling might prevail against the nucleation rate, resulting in centric cell formation, solely at the industrial scale. Another explanation could be the possible cell coalescence depending on low nucleation caused by the relatively low pressure of dissolved gas. Overall, the incorporation of 2 wt % xGnP nanoplatelets subjected to low SCF (0.25 wt %) had a reducing effect on cell size regardless of addition method.

Surprisingly, the presence of xGnP nanoplatelets did not promote cell density for both methods (Table 6). The average cell density of neat samples was proportionally calculated as 8.58 × 10^5^ cm^−3^ and 28.33 × 10^5^ cm^−3^ of SCF loading level. This result is higher by approximately six orders of magnitude than that of the corresponding nanocomposites for each SCF level, regardless of xGnP level and addition methods, even in the presence of a coupling agent. According to the literature, the addition of nanoparticles enhances cell nucleation even at low concentrations. A fourfold increase in cell density was observed in the presence of 3 wt % nanotalc for a polypropylene-based composite prepared via twin-screw extrusion [7]. The number of formed cells in a specific area increased with the SCF amount gradually, whereas the concentration of SCF was the most effective parameter of average cell density in the study. A similar enhancement is proven for foam injection molding of a prefoam nanocomposite [20]. The average skin thickness/foam thickness ratio changed according to the average cell size of samples. The average skin thickness decreased with an increase in SCF % in all samples, as expected. The effect of the graphene nanoplatelet amount had a slight reducing effect on skin thickness–foam thickness ratio. This might be explained by the orientation and dispersion of xGnP particles through the skin layer regarding the dominant shear force because of low SCF.

#### 3.2.2. Mechanical Properties

The flexural modulus, strength, and Izod impact strength of the foam samples are presented in Figure 6, Figure 7 and Figure 8. The flexural modulus was enhanced in the presence of xGnP for all solid nanocomposites (Figure 6).

The nanocomposites with 2 wt % prepared via Method 2 showed the ultimate modulus values. Although the nanocomposite prepared by Method 2 exhibited up to a 30% increase compared to its neat counterpart, the nanocomposite prepared by Method 1 showed an increase of 10.7 and 17.1 orders of magnitude proportional to the xGnP nanoplatelet amount, respectively. The difference proved the effect of the coupling agent in the reinforcing mechanism. The flexural modulus of neat foam samples dramatically reduced with SCF level, whereas the deteriorative effect of foaming was observed for nanocomposites prepared via Method 2. Despite this, the increment on SCF level from 0.25 to 0.35 wt % improved the modulus of samples prepared by Method 1 as a prospect of possible orientation, dispersion, or exfoliation of platelets. As a consequence, the addition of xGnP nanoplatelets compensates for the flexural modulus loss. The flexural strength of samples was prone to similar modulus behavior in terms of the reinforcement role of xGnP nanoplatelets. The findings agree with the literature [20,21]. Nanocomposites with 2 wt % xGnP via Method 2 showed the highest flexural strength (Figure 7).

An approximately twofold increase in flexural strength was obtained for all nanocomposites processed with high SCF levels compared to their neat counterpart. This enhancement can be attributed to the increased crystallinity and alignment of platelets. Consequently, xGnP incorporated foam samples showed enhanced flexural behavior compared to neat solid counterparts. So, the foam samples demonstrating a 12% weight reduction fulfill the requirements of the automotive industry. The impact behavior of all foam samples decreased regarding the notch effect of foam cells (Figure 8). Although neat co-PP solid and foams with low SCF showed the ultimate impact strength among the nanocomposite counterparts, contrarily, the presence of xGnP recovers the loss of impact strength against the neat foam counterpart under the high SCF level conditions. Increased skin thickness provides toughness, and the reinforcement of xGnP plays a more efficient role than that of cellular morphology. Consequently, the foam nanocomposite did not meet the requirements of the automotive industry.

#### 3.2.3. Structural Analysis

In light of the morphological and mechanical characterization, injection molding and SCF effects on the dispersion of xGnP in the co-PP matrix were investigated for both methods via X-ray diffraction (XRD) and Raman spectroscopy. Figure 9a shows that the shoulder peak appeared in 15.8°, and the dramatic decrease in the peak for xGnP at 26.8° showed that processing with even a low SCF level affects the distribution of xGnP for Method 2. The peak at 26.8°, which belongs to the (002) plane of xGnP, was not observed for the neat sample and its nanocomposite counterparts. However, the shoulder peak corresponding to the phase of co-PP had a gradually increased amount of SCF. A similar change was observed for samples prepared by Method 1 with 2 wt % xGnP (Figure 9b).

Additionally, the disappearing of the peak at 2θ = 28.6° is attributed to the full exfoliation of the xGnP layers. For further investigation, the skin layer of samples prepared by Method 1 with 2 wt % xGnP was analyzed via Raman spectroscopy. The result is displayed in Figure 10. The alignment of xGnP samples along the skin layer and the separation degree of interlayers were revealed. The intensity of the peak belonging to G and 2D increased and the peaks became visible (Figure 10).

As intensity depends on the concentration of the xGnP, the alignment of xGnP through skin layers was proven. According to Table 7, the widening of the 2D band is an indicator of the exfoliation degree. Additionally, the shift of the G peak in lower wavenumbers shows the enhancement of exfoliation. Our finding agrees with the literature [44].

#### 3.2.4. Rheological Properties

As mentioned above, rheological properties in terms of melt strength play a critical role in cell growth and stability. Figure 11a–d represent the storage modulus (G′), loss modulus (G″), loss tangent (tan δ), and complex viscosity (η*), respectively, of the prepared samples in the frequency range from 0.1 to 600 rad s^−1^.

The storage modulus of all samples proportionally increased with frequency. Moreover, nanocomposites with 1 wt % behaved almost the same at higher frequencies. Neat samples showed the highest storage compared to their nanocomposites, whereas nanocomposites with 2 wt % xGnP exhibited the lowest. Nanocomposites prepared by Method 2 showed a lower storage modulus than that of the corresponding sample due to the presence of a coupling agent. Likewise, the loss modulus of all samples showed an increase with increased frequency. The lowest loss modulus was obtained in samples with 2 wt % xGnP. As a result, the loss tangent, tan δ, was reduced in the order of Methods 1 and 2, and the neat samples. A decrease in complex viscosity exhibited consistency with the MFI values of nanocomposites. Platelets were prone to sliding over polymer chains rather than restricting mobility via network formation [45]. xGnP with a similar aspect ratio provided network formation polymer chains even in the absence of a coupling agent in the literature [41]. The orientations of platelets driven by melt flow and shear forces might restrict the network formation, and this supports the discussion of the morphological analysis and agrees with the literature. Industrial processing conditions such as large pressure differences regarding long-distance melt flows, high shear stress and the SCF plastifying effect thus play a dominant role, and the structure–property relationship of nanocomposites was hindered under harsh processing conditions for industrial-scale applications such as large parts.

#### 3.2.5. Sound Transmission Loss Property

Transmission loss (TL) is the aspect that relies on bulk modulus as the intrinsic property of the material, and cell density and the distribution of the foam structure.

Figure 12a compares the sound transmission loss results of nonfoam injection molded samples. Neat co-PP showed a general pattern indicating a decline in frequency from 1000 Hz to 2000 Hz, followed by an incremental peak between 2000 and 4000 Hz, and a sharp decrease to 6000 Hz. xGnP nanocomposites showed a similar trend to that of the neat samples. Graphene is the lightest sound-absorbing material within a frequency range from about 60 to 6300 Hz regarding its intrinsic properties, and role as crystallization promoter and modulus [46]. The orientation of platelets with the flow had a significant effect on air flow resistivity [8,47]. According to Figure 12b,c, the addition of xGnP enhances transmission loss. The highest TL value of samples was obtained in the range of 2000–4000 Hz in the sample of compounded co-PP nanocomposites comprising 2 wt % xGnP, whereas the lowest TL value is in the neat sample. As a result, mmco-PP-xGnP2-M1 is a promising material for automotive exterior trim parts.

## 4. Conclusions

This study proposed a practical one-step approach via diluting a graphene nanoplatelet (xGnP) masterbatch as a process aid to enhance the weight reduction and mechanical properties of copolymer polypropylene (co-PP)-based foam automotive trim parts. According to our results, the incorporation of xGnP enhances the flowability, mechanical and thermal properties, and foaming behavior of co-PP. The presence of xGnP reduces the nonuniformity and polydispersity of the material. It contributes to the increase in the skin-foam thickness ratio of the foam and the elimination of cell coalescence. Increased cell size and decreased cell density were achieved in all samples comprising 2 wt % xGnP. However, the presence of a coupling agent and possible chain scission in twin-screw extrusion steps exhibited grainier cell sizes compared to those of the proposed method in this study. Flexural properties were improved with the addition of 2 wt % xGnP for both methods. However, the impact strength of the samples needs optimization. Increased supercritical fluid (SCF) levels improved flexural behavior in Method 1 regarding the orientation of particles and the exfoliation of layers. The sound transmission loss behavior of the nanocomposites was improved by up to 40 dB regarding the enhanced modulus for frequencies at which the bulk modulus plays a more important role than that of thickness. The foam of the 2 wt % xGnP nanocomposite prepared via Method 1 with high SCF enables improved sound transmission loss in lower frequencies compared to neat foams.

Consequently, the proposed method provided postulated enhancement on the cell formation, and mechanical and sound insulation performance of the foams by the addition of xGnP with 2 wt %. Method 1 with a high level of SCF was attained as the optimal process for the nanocomposites in this study. Additionally, processing the 12% weight-reduced part was enabled without any shot-shot or postblowing problems. The flow-oriented nanoplatelet structured morphology provides an opportunity for applications relying on enhanced thermal and electrical conductivity. Therefore, the proposed method has potential for electromagnetic interference shielding and the elimination of metal thermal dissipater subparts of engine cover parts. The approach seems feasible for manufacturing plastic parts such as luggage trims and engine undertrims in automotive applications.

## Figures and Tables

**Figure 1 polymers-14-01192-f001:**
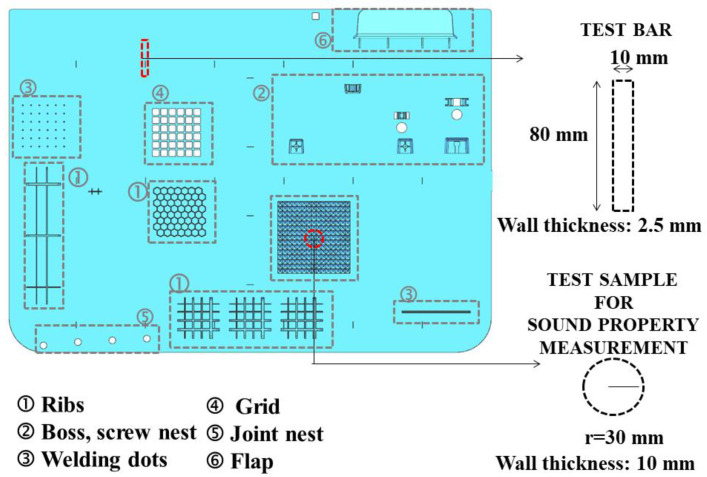
Experimental part design with elements of (1) ribs, (2) boss-screw nests, (3) welding dots, (4) grids, (5) joint nests, (6) flaps, and sound transmission loss. Locations of standard ISO test specimens for mechanical, scanning electron microscopy, and sound transmission loss measurements marked with dashed red line.

**Figure 2 polymers-14-01192-f002:**
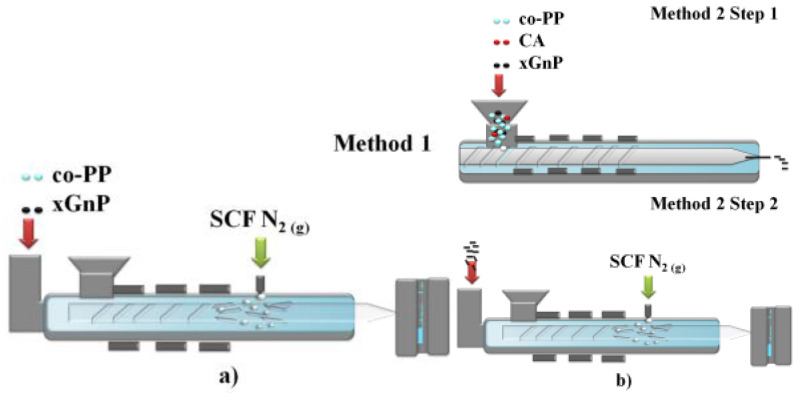
Schematic representation of (**a**) Method 1 and (**b**) Method 2.

**Figure 3 polymers-14-01192-f003:**
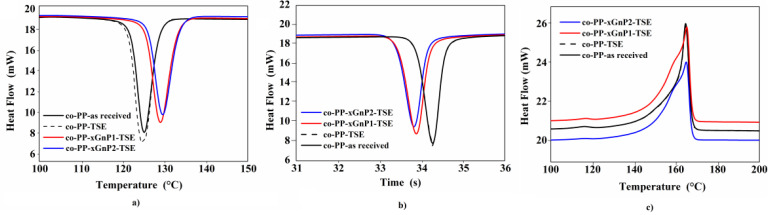
DSC thermograms of crystalization and melt curves of neat co-PP (as received and its composites with 1 and 2 wt % xGnP samples): (**a**) crystallization curve in first cooling cycle of samples; (**b**) crystallization time curve in the first cooling cycle of samples; (**c**) melting curves in the second heating cycle of samples. DSC: differential scanning calorimetry; co-PP: polypropylene copolymer; xGnP: graphene nanoplatelet.

**Figure 4 polymers-14-01192-f004:**
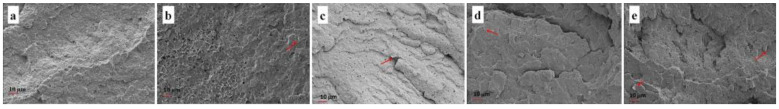
SEM images with magnification of 100 of nonfoamed samples of (**a**) as received neat co-PP; (**b**) 1 wt % xGnP incorporated nanocomposites via Method 1; (**c**) 2 wt % xGnP incorporated nanocomposites via Method 1; (**d**) 1 wt % xGnP incorporated nanocomposites via Method 2; (**e**) 2 wt % xGnP incorporated nanocomposites via Method 2. SEM: scanning electron microscopy, xGnP: graphene platelet.

**Figure 5 polymers-14-01192-f005:**
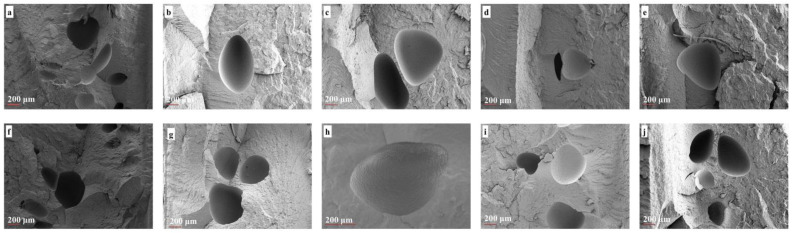
SEM images with magnification of 200 of (**a**) neat co-PP foam; (**b**) 1 wt % xGnP incorporated by Method 1; (**c**) 2 wt % xGnP incorporated by Method 1; (**d**) 1 wt % xGnP incorporated by Method 2; (**e**) 2 wt % xGnP incorporated by Method 2, subjected to low SCF level (0.25 wt %) and (**f**) neat co-PP foam; (**g**) 1 wt % xGnP incorporated by Method 1; (**h**) 2 wt % xGnP incorporated by Method 1; (**i**) 1 wt % xGnP incorporated by Method 2; (**j**) 2 wt % xGnP incorporated by Method 2, subjected to a high SCF level (0.35 wt %). SEM: scanning electron microscopy; xGnP: graphene platelet; co-PP: copolymer polypropylene; SCF: supercritical fluid.

**Figure 6 polymers-14-01192-f006:**
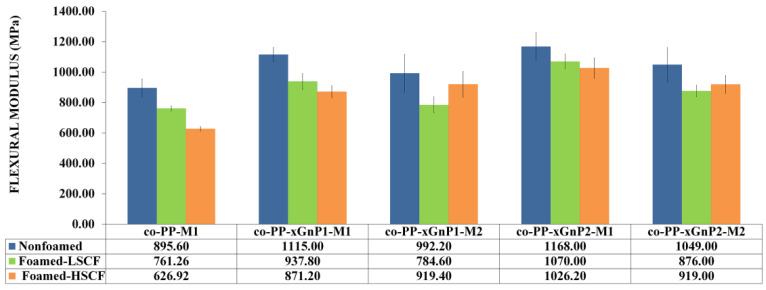
Flexural modulus of samples.

**Figure 7 polymers-14-01192-f007:**
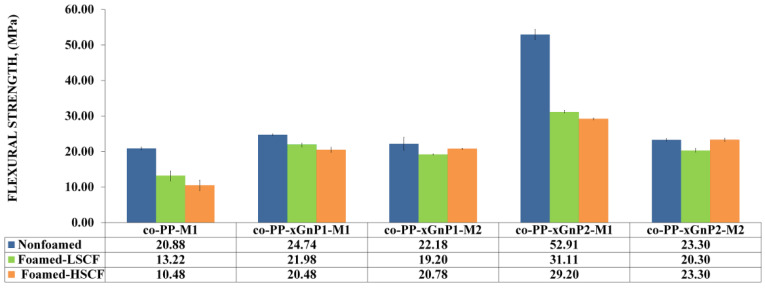
Flexural strength of samples.

**Figure 8 polymers-14-01192-f008:**
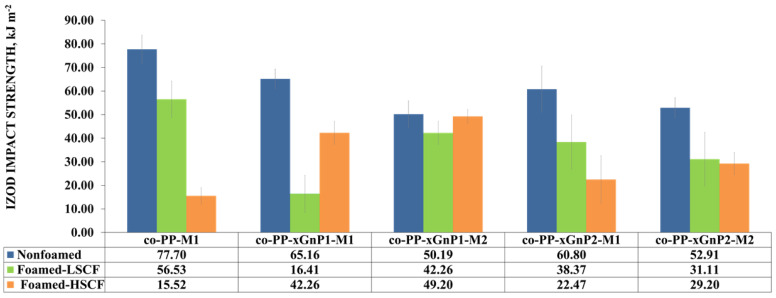
Izod impact strength of samples.

**Figure 9 polymers-14-01192-f009:**
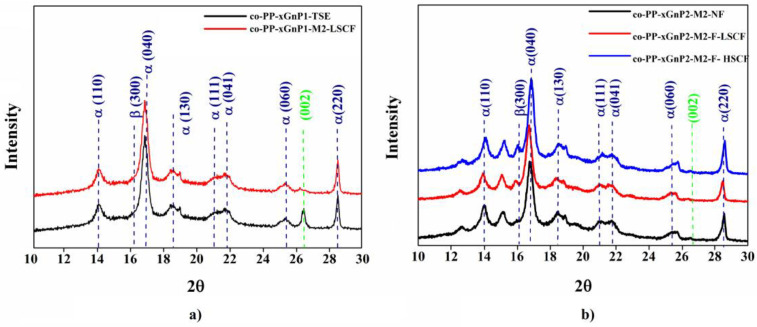
Comparative diffraction pattern of representative nanocomposites. (**a**) Neat and foamed samples with 1 wt % xGnP via Method 1 with low SCF amount; (**b**) neat and foamed sample with 2 wt % xGnP via Method 2 with low and high SCF amount. xGnP: graphene platelet.

**Figure 10 polymers-14-01192-f010:**
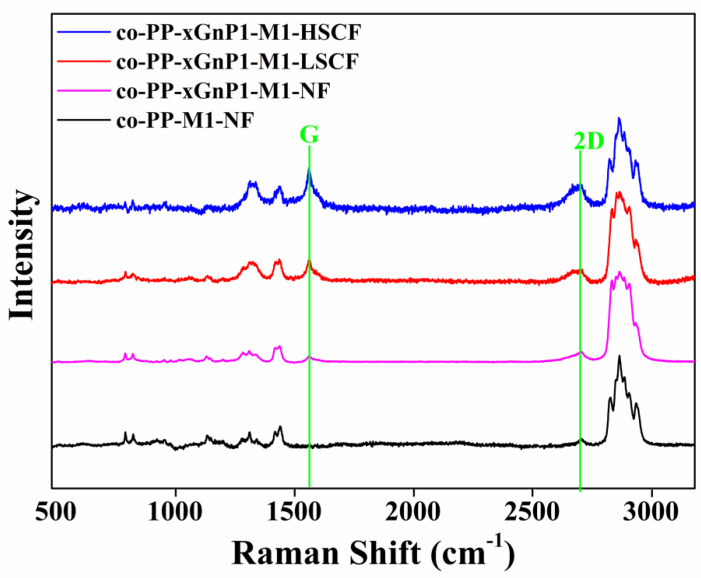
Raman spectrum of nanocomposite with 1 wt % xGnP prepared by Method 1. xGnP: graphene nanoplatelet.

**Figure 11 polymers-14-01192-f011:**
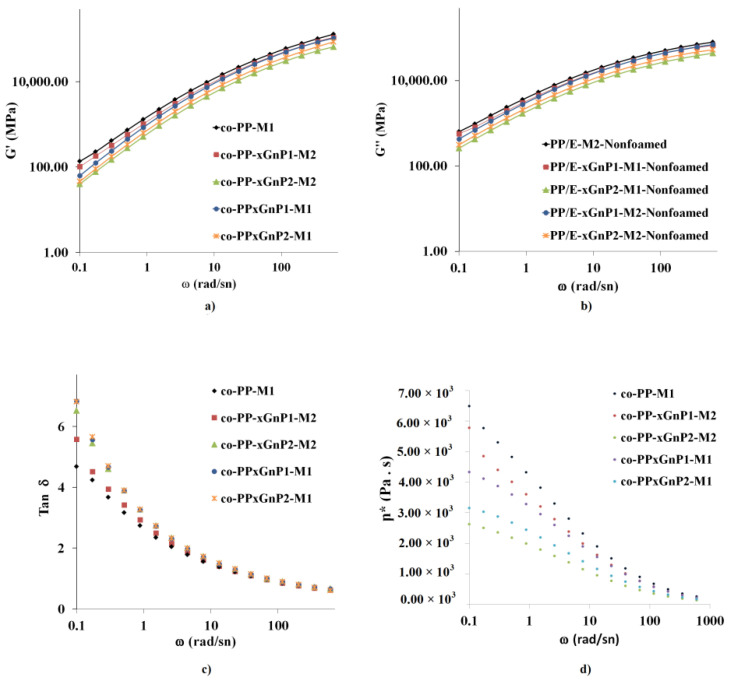
Rheological measurement results of (**a**) storage modulus, (**b**) loss modulus, (**c**) tan δ, and (**d**) complex viscosity of prepared samples.

**Figure 12 polymers-14-01192-f012:**
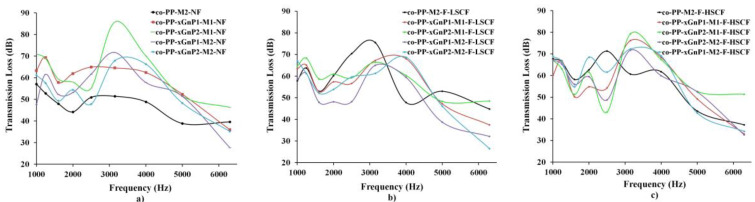
Sound transmission loss results of (**a**) nonfoam (solid), (**b**) foam with a low SCF amount and (**c**) foam with a high SCF amount. SCF: supercritical fluid.

**Table 1 polymers-14-01192-t001:** Foam injection molding process parameters.

Process Parameters	Nonfoam (Solid) Injection Molding	Foam Injection Molding
Injection speed (mm s^−1^)	50–40	110–120
Injection pressure (bar)	1200–1100	1000–1000
Back pressure (bar)	90	90
Screw stroke (mm)	146	135
Clamping pressure (bar)	250	150
Clamping time (s)	40	0.2
Clamping stroke (mm)	20	8
SCF pressure (bar)	0	100
SCF amount (%)	-	0.25–0.35
SCF barrel residence time (s)	-	20
Total cycle time (s)	60	42

**Table 2 polymers-14-01192-t002:** Prepared samples by twin-screw extrusion (TSE) as Step 2 of Method 2.

Sample Code	xGnP (%)	PP-g-MAH (%)
co-PP-TSE	0	1
co-PP- xGnP1-TSE	1	1
co-PP -xGnP2-TSE	2	1

TSE: twin-screw extrusion; xGnP: graphene nanoplatelet; PP-g-MAH: maleic anhydride grafted polypropylene; co-PP: copolymer polypropylene.

**Table 3 polymers-14-01192-t003:** Prepared foam injection-molded samples via Method 1 (M1) and Method 2 (M2).

	Sample Code	Method	xGnP (%)	PP-g-MAH (%)	SCF (%)
Non-Foam (NF)	co-PP-M1-NF	Method 1	0	0	0
	co-PP-xGnP1-M1-NF	Method 1	1	0	0
	co-PP-xGnP2-M1-NF	Method 1	2	0	0
	co-PP-xGnP1-M2-NF	Method 2	1	1	0
	co-PP-xGnP2-M2-NF	Method 2	2	1	0
Foam- with Low SCF (LSCF)	co-PP-M1- F-LSF	Method 1	0	0	0.25
	co-PP-xGnP1-M1-F-LSCF	Method 1	1	0	0.25
	co-PP-xGnP2-M1-F-LSCF	Method 1	2	0	0.25
	co-PP-xGnP1-M2-F-LSCF	Method 2	1	1	0.25
	co-PP-xGnP2-M2-F-LSCF	Method 2	2	1	0.25
Foam with High SCF (HSCF)	co-PP-M1-F-HSCF	Method 1	0	0	0.35
	co-PP-xGnP1-M1-F-HSCF	Method 1	1	0	0.35
	co-PP-xGnP2-M1-F-HSCF	Method 1	2	0	0.35
	co-PP-xGnP1-M2-F-HSCF	Method 2	1	1	0.35
	co-PP-xGnP2-M2-F-HSCF	Method 2	2	1	0.35

xGnP: graphene nanoplatelet; PP-g-MAH: maleic anhydride grafted polypropylene. SCF: Super ciritical fluid.

**Table 4 polymers-14-01192-t004:** Physical, mechanical, and thermal properties of the prepared semiproducts.

Property	xGnP MB	co-PP (as Received)	co-PP-TSE	co-PP-xGnP1-TSE	co-PP-xGnP2-TSE
xGnP, (wt %)	10.0	0.00	0.00	1.00	2.00
Density (gr cm^−^³)	0.94	0.90	0.90	0.90	0.91
Melt flow index (g 10 min^−1^)	14.90	14.32	15.6	16.9	18.4
Ash content (%)	0.00	0.00	0.00	0.68	0.89
Tensile modulus (MPa)	2200.00 ± 72.00	1110.00 ± 27.00	998.00 ± 18.00	1270.00 ± 38.00	1380.00 ± 56.00
Tensile strength (MPa)	36.60 ± 0.56	17.90 ± 0.42	16.60 ± 0.36	20.60 ± 0.36	22.20 ± 0.72
Flexural modulus (MPa)	2200.00 ± 103.00	950.00 ± 28.00	902.00 ± 56.00	1100.00 ± 58.00	1220.00 ± 48.00
Flexural strength (MPa)	54.70 ± 0.56	26.60 ± 0.53	25.30 ± 0.51	30.00 ± 0.47	32.50 ± 0.26
Izod impact strength (kJ m^−2^)	1.73 ± 0.05	19.4 ± 0.06	18.3 ± 0.01	15.17 ± 0.06	10.23 ± 0.09
Heat deflection Temperature (°C)	-	100.3 ± 0.13	70.03 ± 0.12	81.02 ± 0.13	81.58 ± 0.21
Vicat softening point (°C)	-	67.31 ± 0.14	67.24 ± 0.23	68.68 ± 0.12	70.78 ± 0.31

xGnP MB:graphene nanoplatelet masterbatch; co-PP: copolymer polypropylene; TSE: twin screw extrusion.

**Table 5 polymers-14-01192-t005:** Summary of the thermal parameters of neat samples and xGnP nanocomposites extracted from DSC measurements.

Samples	T_c_ (°C)	−ΔH_c_ (J g^−1^)	T_m_ (°C)	ΔH_m_ (J g^−1^)
co-PP (as received)	125.21	64.90	164.67	52.56
co-PP-TSE	124.93	63.61	163.52	50.89
co-PP-xGnP1-TSE	129.11	65.68	165.25	58.92
co-PP-xGnP2-TSE	129.68	66.02	164.94	61.79

DSC: differential scanning calorimetry; Tc: crystallization temperature; ΔHc: enthalpy of crystallization; Tm: melting temperature; ΔHm: enthalpy of melting.

**Table 6 polymers-14-01192-t006:** Quantitative cell data derived from SEM images of neat PP copolymer and its 1–2 wt % xGnP nanocomposites.

Sample Code		xGnP (%)	SCF (%)	Weight Reduction Ratio (%)	Avr. Cell Density × 10^5^ (Number/cm^3^)	Avr. Cell Size (µm)	Avr. Skin Thickness/Foam Thickness Ratio
Foam—Low SCF	co-PP-M1	0	0.25	7	8.58	470.1	1.43:1
	co-PP-xGnP1-M1	1	0.25	7	1.87	500.02	2.42:1
	co-PP-xGnP1-M2	1	0.25	7	1.80	707.5	1.28:1
	co-PP-xGnP2-M1	2	0.25	7	1.82	243.7	2.63:1
	co-PP-xGnP2-M2	2	0.25	7	1.91	370.9	1.29:1
Foam—High SCF	co-PP-M1	0	0.35	12	28.33	366.5	0.60:1
	co-PP-xGnP1-M1	1	0.35	12	5.00	424.36	1.54:1
	co-PP-xGnP1-M2	1	0.35	12	4.82	537.64	1.49:1
	co-PP-xGnP2-M1	2	0.35	12	4.77	384.45	1.28:1
	co-PP-xGnP2-M2	2	0.35	12	4.95	384.61	1.41:1

xGnP: graphene nanoplatelet; SCF: supercritical fluid; avr.: average; co-PP: copolymer polypropylene; M1: method 1, M2: method2.

**Table 7 polymers-14-01192-t007:** The data extracted from the Raman spectrum of samples.

Sample	2D Intensity	2D FWHM	Peak Position
Co-PP-M2-NF	1128.99	42.98	2723
Co-PP-xGnP2-M2-NF	1533.57	67.09	2712.22
Co-PP-xGnP2-M1-LSCF	2542.41	81.55	2703.40
Co-PP-xGnP2-M1-HSCF	4561.31	89.01	2699

xGnP: graphene nanoplatelet; NF:non-foamed; avr.:average; co-PP: copolymer polypropylene; M1:method 1. M2: method2. LSCF: low level super ciritical fluid. HSCF: high level super critical fluid. FHWM: full width half maxima

## Data Availability

The data presented in this study are available on request from the corresponding author. The data are not publicly available due to privacy.

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
