# Peer review of "An Industrial Case for Polypropylene Nanocomposite Foams: Lightweight, Soundproof Exterior Automotive Parts"

_polymers, 2022, doi:10.3390/polym14061192_

Round 1

Reviewer 1 Report

Özünlü and Guner in this work report an industrial-scale approach to fabricate polypropylene foams aiming to automotive applications. The method involves the incorporation of graphene nanoplatelets and PP-g-MAH in co-PP. The proposed method has potential for EMI shielding and the elimination of metal thermal dissipater sub-parts of engine cover parts. My comments are as follows.

  1. This work is lengthy and shows many data. However, the main effects of the graphene nanoplatelets and the PP-g-MAH on the properties and foaming of co-PP are not clearly explained and pointed out. I suggest the key effects should be included in the abstract and conclusions. Otherwise the manuscript is too scattered and difficult to follow.
  2. I don’t think “the melt strength reduced as loss tanδ decreased” (Page 15). Please check the validity of this argument. Since the xGnP decreases the viscosity of the co-PP, though not proportionally related, the melt strength of co-PP should decrease as xGnP is incorporated.
  3. The abbreviation such as SCF and VST should be defined when first time appear in the article, even in the abstract.
  4. In Figure 4, the labels of the images are not consistent with the caption.
  5. Line 477 on Page 12, check the cell density values.

Author Response

Dear Prof,

Thank you for your report. We appreciate the time and effort that you dedicated to providing feedback on our manuscript and are grateful for the insightful comments on and valuable improvements to our paper.
We have incorporated the changes and point-by-point responses in the attachment. 

Sincerely,

Reviewer 2 Report

 Özünlü and Guner paper entitled on “A Facile, Cost-Effective Industrial-Scale Approach to Fabrication of Lightweight, Soundproof Polypropylene Nanocomposite Foams for Automotive Applications” has been reviewed thoroughly. The authors presented the work in a systematic manner. The authors showed various evidence to enhance the properties such as thermal and mechanical of co-polypropylene nanocomposite foam with the addition of nano-platelets. It is one of the important research topics in the automotive industry. I recommend this work accepted in this journal after addressing the following comments carefully.

Specific comments are as follows,

  • The authors should change the paper title first. It should be short and clear.
  • Make sure all abbreviations are written out in full the first time when they are used. This is particularly important in the Abstract and in the Conclusions sections. When first mention, please use the full name, not the abbreviation form like xGnP-MB, SCF etc., in abstract. Please check this in the manuscript.
  • Please provide space between percentage and x for example 1-2 wt%xGnP written as 1-2 wt% xGnP. Check this throughout the manuscript.
  • Similarly, xGnP-coPP should be written as xGnP-co-PP
  • Maleic a n hydride written as Maleic an hydride in the section 2.1 and 10 °C min-1 is written as 10 °C min-1.
  • izod unnotched should written as Izod unnotched
  • The Figure 1, 6, 7, 8, 11, 12 is not visible to the reader. Please change it.
  • Provide space between number and unit (190ºC to 190 ºC) and author should use appropriate degree Celsius Sign. Check this throughout the manuscript.
  • Density formula given in section 2.3.2 is not appropriate format. Please change it according to the equation format.
  • Avoid abbreviation in the heading and subheading.
  • Check language and grammar thoroughly in the manuscript.
  • Please indicate magnification ranges on SEM images.
  • What is full form of EMI? Please use the full form in conclusion parts.
  • Include important finding in the conclusions part
  • Check all the references, and the format of the references should be uniform and same as in materials journal format.
  • Authors requested to cite very recent references related to this work.

Author Response

(The authors gave the same response as above.)

Reviewer 3 Report

The article presents “a new practical approach to regulate the cell formation of copolymer polypropylene (co-PP) by means of the utilization of graphene nanoplatelets (xGnP) as a process aid during foam injection molding” with the view of application in the automotive industry. The amount of xGnP is studied, as well as two different levels of supercritical fluid (SCF) and the samples were compared with those prepared by a traditional method as concerning their mechanical and acoustical performance The article presents novelty, is fitting the journal scope and is well organized in terms of experiments and discussion. The English undergone editing by MDPI and is OK. However, there are several errors, not only typos, which occurred during writing and introducing the text and figures into the template, which indicates the lack of attention. These aspects raise high concern.

Some suggestions for authors are given bellow in order to improve the manuscript:

Line 2: The title mentions “cost-effective” however there is no discussion about this aspect all over the text, hence is suggested to remove.

Abstract, Line 15 and 16: The authors should explain what “MB” and “SCF” are, because is the first time when using the abbreviation.

Line 126: A typo occurred, and the authors should remove the space between “a” and “nhydride”, should be “anhydride”.

Line 129-130: The authors state the content of maleic anhydride was determined by FT-IR, how is this possible, since this technique is qualitative, namely for identification of chemical bonds. Can you detail, please?

Line 138: Subsection 2.2.1 is not necessary, since no 2.2.2 exists….

Figure 1: The quality of Figure 1 is not very good; this should be improved.

Line 150: In the caption of Figure 1, the “and” should be placed before “(7).

Line 158: Should be “(Figure 2b)” after “Method 2”, in line with what was stated before.

Line 163-165: The caption of Figure 2 can be simplified as “Schematic representation of (a) Method 1 and (b) Method 2…” to avoid repetition.

Figure 2: Is not explained what “CA” represent…”coupling agent”? Maybe introduce it at line 17, where is first mentioned…

Line 167: “.” after “wt%” should be deleted.

Line 171-172: What was the criteria to choose the range of temperature? Is not obeying any rule: increase from 5 to 5 ºC, or 10 to 10 ºC?! additionally 220 ºC is repeated.

Table 3: The content of this table should be maintained all on the same page.

Line 230: Please verify superscript in “min-1”.

Line 262: A typo occurred, please delete “0” after “achieve”.

Line 253-255: What is the difference between “stretching speed” and “test speed”, why are they different? Different methods are used? Is supposed to be a tensile test.

Table 4: Please, try to keep all table on the same page, furthermore an error occurred when introducing this table, the line numbers were moved to the left side, overlapping the Table. Please, check this.

Figure 3a: Inside the Figure 3a the blue line should be co-PP-xGnP2-TSE.

Line 363, Figure 3: In the caption, please introduce a space between “1” and “and”.

Figure 4: The Figures are inverted, since the scale numbers and letters are upside down.

Line 421, Figure 4: In the caption is missing “;” before “b)”

Line 443: Caption of figure 5, please correct “la ow” with “low”.

Table 6: several errors occurred in this table: i) in the last column, the values in the last 4 rows are wrong (they were copied from the previous column), ii) in the average cell size column the numbers should be represented with the same significative figures.

Line 477-478: An error occurred when presenting the average cell density values “8.58x105” and “28.33x105”, this should be corrected.

Line 544-545: Caption of Figure 9 states that in a) is presented method 1, however inside this figure “co-PP-xGnP1-M2-LSCF” is present. The authors should correct this.

Line 553: The caption of Figure 10 is wrong, since it presents nanocomposites with 2 wt% xGnP, but inside the figure ”GnP1” is present.

Figure 11 is too small, this has to be increased since is hardly visible.

Line 594: In figure 12 caption, “and” should be placed before c)”

Figure 12 is small and the legend inside is hardly visible; the letters

General comment: In all figures there are abbreviations that are explain, which are not necessary, since they were used before.

Author Response

(The authors gave the same response as above.)

Round 2

Reviewer 1 Report

I recommend publication.

Reviewer 3 Report

The authors answered the questions addressed by the reviewers and improved the text and figures of the manuscript.